# FINDING DEEP LOCAL OPTIMA USING NETWORK PRUNING

## ABSTRACT

Artificial neural networks (ANNs) are very popular nowadays and offer reliable solutions to many classification problems. However, training deep neural networks (DNN) is time-consuming due to the large number of parameters. Recent research indicates that these DNNs might be over-parameterized and different solutions have been proposed to reduce the complexity both in the number of parameters and in the training time of the neural networks. Furthermore, some researchers argue that after reducing the neural network complexity via connection pruning, the remaining weights are irrelevant and retraining the sub-network would obtain a comparable accuracy with the original one. This may hold true in most vision problems where we always enjoy a large number of training samples and research indicates that most local optima of the convolutional neural networks may be equivalent. However, in non-vision sparse datasets, especially with many irrelevant features where a standard neural network would overfit, this might not be the case and there might be many non-equivalent local optima. This paper presents empirical evidence for these statements and an empirical study of the learnability of neural networks (NNs) on some challenging non-linear real and simulated data with irrelevant variables. Our simulation experiments indicate that the cross-entropy loss function on XOR-like data has many local optima, and the number of local optima grows exponentially with the number of irrelevant variables. We also introduce a connection pruning method to improve the capability of NNs to find a deep local minimum even when there are irrelevant variables. Furthermore, the performance of the discovered sparse sub-network degrades considerably either by retraining from scratch or the corresponding original initialization, due to the existence of many bad optima around. Finally, we will show that the performance of neural networks for real-world experiments on sparse datasets can be recovered or even improved by discovering a good sub-network architecture via connection pruning.

## 1 INTRODUCTION

Artificial neural networks are very popular nowadays and they can offer good solutions to many vision problems (Dahl et al., 2013; Krizhevsky et al., 2012; Le, 2013; He et al., 2016). However, the neural networks usually require massive amounts of training data for good generalization, which limit their applications to non-vision problems where the data is not so abundant.

In this paper, we conduct an empirical investigation of the relationship between the amount of training data and the landscape of the loss function. We will see that when the training data is large, the landscape has fewer local optima, which makes it is easier to find a deep local optimum, corresponding to a model with good generalization. If the training data is limited, the energy landscape could have in some cases so many local minima that it is computationally prohibitive to find a deep local minimum with good generalization.

For this experimental investigation, we will use it as a proxy of a difficult non-linear dataset generalized XOR data with extra irrelevant variables. By adjusting the dimensionality of the XOR relationship and the number of irrelevant variables we can control the degree of difficulty of the problem and see the relationship between the amount of training data and the loss landscape.

For our experimental investigation, we will focus our attention to simple neural networks with one hidden layer and ReLU activation. On the XOR-based classification problems, we will show that the cross-entropy energy landscape has many shallow local minima and some deep minima that are very hard to find. We will also observe that the number of shallow local minima grows prohibitively large as the number of irrelevant variables increase, and the chances of finding a deep minimum decrease quickly with the number of irrelevant variables. This decrease is much faster for harder problems (e.g. 5D XOR) than for easier problems (3D XOR). We also observe that for a fixed problem, the difficulty of finding deep local minimum decreases as more training examples are used.

Based on these findings, we propose a graduate node and connection pruning technique that starts with a dense network with many hidden nodes and gradually prunes connections or nodes to arrive at a sparse network with few hidden nodes. A large number of initially hidden nodes ensure high connectivity between the local minima making them easier to find. The gradual pruning ensures that the local optimum is not lost during the optimization process. We will see in experiments that this technique finds deeper local optima than when starting from a small network with random initialization or retrain the found sparse sub-network from scratch or using corresponding original initial parameters.

This is a purely experimental paper that does not intend to make any theoretical contributions. Through some well-designed experiments on data of different levels of difficulty it tries to obtain insights about the relationship between data difficulty, the number of training examples and the computational difficulty of finding a deep local optimum with good generalization. This intuition suggests that in some cases a neural network could be trained with fewer training examples at the expense of increased computational complexity in the optimization process of finding a deep local optimum, where a simple SGD or Adam optimization might not be enough.

## 1.1 RELATED WORK

**Local minima.** Recent studies Draxler et al. (2018); Garipov et al. (2018) have shown that the local minima of some convolutional neural networks are equivalent in the sense that they have the same loss (energy) value and a path can be found between the local minima along which the energy stays the same. For this reason, we will focus our attention to fully connected neural networks and find examples where the local minima have different loss values. Moreover, Soudry & Carmon (2016) proves that all differentiable local minima are global minima for the one hidden layer NNs with piecewise linear activation and square loss. However, nothing is proved for non-differentiable local minima.

**Network pruning.** There has been quite a lot of work recently about neural network pruning, either for the purpose of improving speed and reducing complexity or giving insights about explaining the essential capability of the pruning technique. Han et al. (2015b) and Han et al. (2015a) propose the "Deep Compression", a three-stage technique, which significantly reduces the storage requirement for training deep neural networks without affecting their accuracy. Liu et al. (2018) shows that for structured pruning methods, directly training the small target sub-network or pruned model with random initialization can achieve a comparable or even better performance than retraining using the remaining parameters after pruning. They also obtain similar results towards to a unstructured pruning method Han et al. (2015b) after fine-tuning the pruned sub-network on small-scale datasets. Frankle & Carbin (2019) introduces the Lottery Tickets Hypothesis which claims that a random-initialized dense neural network contains a sub-network that can be trained in isolation with the corresponding original initialized parameters to obtain the same test accuracy of the original network after training for the same number of iterations.

## 2 AN EMPIRICAL STUDY OF THE NEURAL NETWORK LOCAL MINIMA

To study the local minima of neural networks, we will look at an extreme case, the XOR problem. The $k$-dimensional XOR is a binary classification problem that can be formulated as

$$y(\mathbf{x}) = I(\prod_{i=1}^{k} x_i > 0), \forall \mathbf{x} \in \mathbb{R}^p$$

Observe that in this formulation the XOR data is $p$ dimensional but the degree of interaction is $k$-dimensional, with $k \leq p$. We call this data the $k$-D XOR in $p$ dimensions. In this paper we will work with $k \in \{3, 4, 5\}$, as $k = 2$ is very simple. We assume that $\mathbf{x} \in \mathbb{R}^p$ is sampled uniformly from $[-1, 1]^p$. The XOR problem an example of data that can only be modeled by using higher order feature interactions, and for which lower order marginal models have no discrimination power. This makes it very difficult to detect what features are relevant for predicting the response $y$.

The neural networks (NN) that we will study are two layer neural networks with ReLU activation for the hidden layer. These NNs can model the XOR data in case $p = k = \{3, 4, 5\}$ very well given sufficiently many hidden nodes.

## 2.1 THE ENERGY LANDSCAPE OF THE NN ON THE XOR DATA

To see the difference between the values of the various local minima, we ran the NN starting with 100 random initializations and plotted in Figures 1 and 2 the values of the local minima sorted in increasing order for $k = 4$ and $k = 5$. We repeated this process 10 times and also plotted the average of the sorted local minima.

From Figures 1 and 2 we observe that the local minima are not equivalent, with the lowest being much smaller than the highest. We also see that for small $p$, the 500 hidden nodes NN has the lowest loss values, but for $p = 100$ the 20 hidden nodes NN obtains the lowest loss.

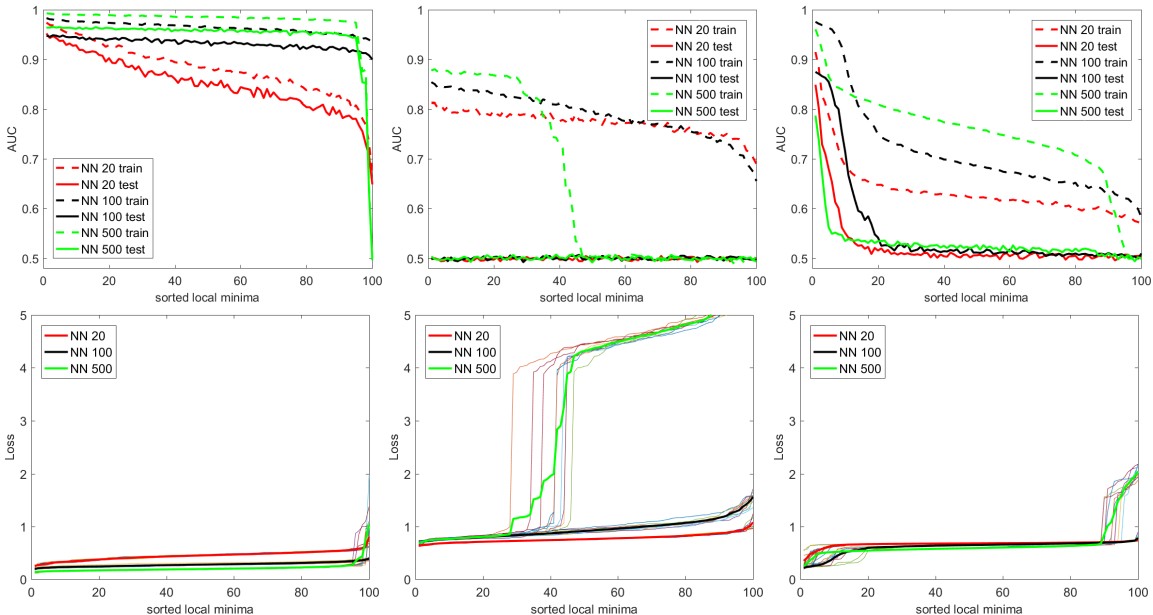

Figure 1: Values of sorted local minima (top) and train and test AUC (bottom) for 4D XOR. Left: $n = 1000, p = 4$. Middle: $n = 3000, p = 16$. Right: $n = 3000, p = 100$.

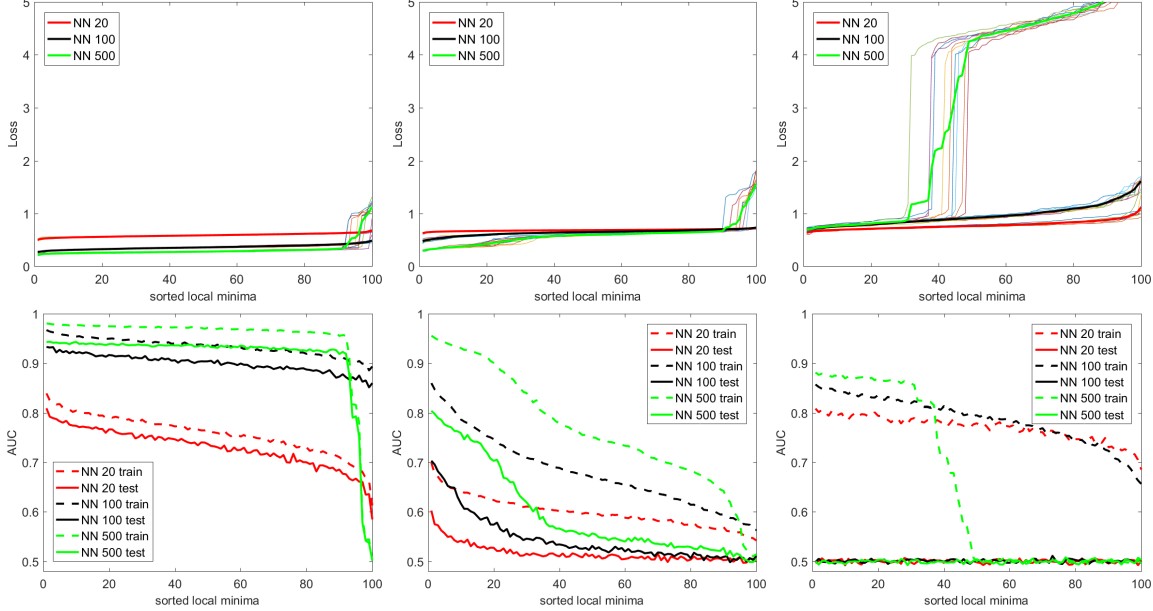

Figure 2: Values of sorted local minima(top) and train and test AUC (bottom) for 5D XOR. Left: $n = 3000, p = 5$. Middle: $n = 3000, p = 10$. Right: $n = 3000, p = 100$.

The loss values are directly related to the training AUC (area under the ROC curve). In Figures 1 and 2, bottom, are plotted the mean train and test AUC of the models corresponding to the sorted local minima. Observe that for $p = 100$ the NN cannot find the global optimum and the test AUCs are all close to 0.5. These experiments indicate that when the number of irrelevant features is small, it is quite easy to find a deep local optimum with good generalization, so the NN is easily trainable. As the number of irrelevant features increases, it becomes harder and harder to find such a deep optimum, and when there are too many irrelevant features it is close to impossible to find a deep optimum with good generalization.

## 2.2 THE TRAINABILITY OF NEURAL NETWORKS

To see the change from an easily trainable NN (such as shown in Figures 1 and 2, left, to a poorly trained NN such as in Figures 1 and 2, right, for each dimension $p$ we train a NN with 500 hidden nodes with 10 random initializations

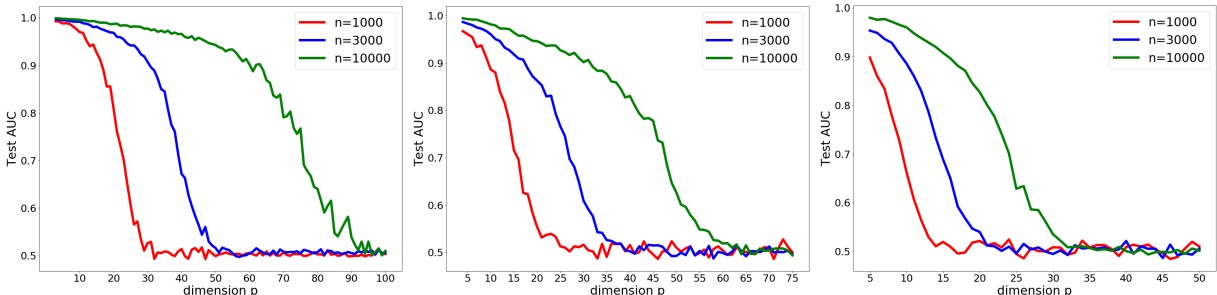

Figure 3: Test AUC of best energy minimum out of 10 random initializations vs. data dimension $p$ for a NN with 500 hidden nodes. Left: $k = 3$. Middle: $k = 4$. Right: $k = 5$.

and keep the solution with smallest loss. Then we compute the test AUC of the obtained NN. For each $p$ we repeat this process 10 times and display in Figure 3 the average test AUC vs $p$.

We observe that the test AUC quickly drops from close to 1 to 0.5. The number of variables $p$ where the test AUC gets below a threshold (e.g. 0.8) depends on the number $n$ of training examples. This drop does resemble a phase transition, from an "easy to train" regime where the local minima are easy to find, to the "hard to train" regime.

Our conclusions from this empirical study are the following:

- If the training data is difficult (such as the XOR data), not all local minima are equivalent, since in Figures 1 and 2 there was a large difference between the test AUC of the best local minimum and the worst one.
- For a fixed training size $n$, the number of shallow local minima quickly blows up as the number of irrelevant variables increases and finding the deep local minima becomes extremely hard.
- If the number of irrelevant variables is not too large (e.g. the middle plot from Figure 2), an NN with a sufficiently large number of hidden nodes will find a deep optimum more often than one with a small number of hidden nodes.

These conclusions form the basis for the proposed connection pruning methodology presented in the next section.

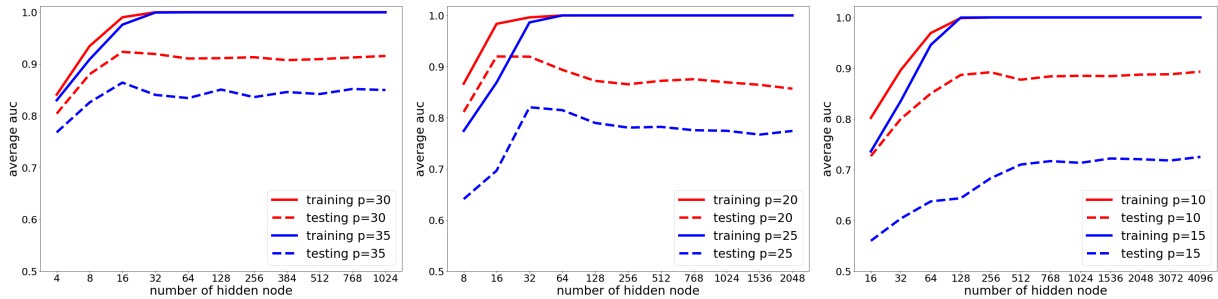

Figure 4: Training and testing AUC vs number of hidden nodes. Left: $k = 3, p = \{30, 35\}$, middle: $k = 4, p = \{20, 25\}$, right $k = 5, p = \{10, 15\}$.

## 3 FINDING BETTER LOCAL MINIMA THROUGH NODE AND CONNECTION PRUNING

As we saw in the previous section, the NNs can handle the XOR data if $p$ is in a reasonable range. We also observe that increasing the number of hidden node in NNs may not be very helpful for improving the test AUC. To demonstrate this observation, for different number of hidden nodes, we train a NN with 10 random initializations and keep the best test AUC and its associated training AUC among the 10 trials. We repeat this process 10 times and display in Figure 4 the average test and train AUC vs the number of hidden nodes. When the number of hidden node increases, the training AUC becomes better and better and finally reaches 1. But the test AUC quickly reaches the best results when the number of hidden nodes is relatively small, and then no further improvement happens as the number of hidden nodes increases. This tells us that increasing the number of hidden nodes, will add too many irrelevant nodes to the NNs, leading to overfitting.

We will use fully connected neural networks with one input layer, several hidden layers ($l = 1, 2, ..., L - 1$) with ReLU activation and one output layer. If the hidden layer $l$ has $h^l$ neurons, we can represent the weights of the hidden nodes in layer $l$ as a matrix $\mathbf{W}^l \in \mathbb{R}^{h^l \times h^{l-1}}$, the biases as a vector $\mathbf{b}^l = (b_1^l, ..., b_h^l)$. Denoting the ReLU activation as

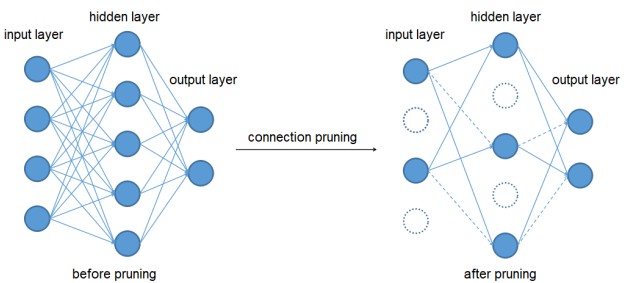

Figure 5: Connection pruning framework for a fully connected NN with one or several hidden layer(s) and one output layer. The input layer is not a real layer with parameters, it consists of the data features which feed into the NNs for training or testing purpose.

$\sigma(x) = \max(0, x)$ we can write the neural network as:
$$f(\mathbf{x}) = \mathbf{W}^L \sigma(\mathbf{W}^{l-1} \sigma(...\sigma(\mathbf{W}^1 \mathbf{x} + \mathbf{b}^1) + ...) + \mathbf{b}^{L-1}) + \mathbf{b}^L \tag{1}$$

### 3.1 CONNECTION PRUNING WITH ANNEALING FOR FEATURE AND NODE SELECTION

The above study showed how important it is to find deep local optima when training neural networks on difficult data with a small number of observations. For that we will start with a larger network which has better connectivity between the local optima, and gradually remove the irrelevant hidden nodes and irrelevant features through connection pruning. Inspired by the feature selection with annealing (Barbu et al., 2017) and the weight learning for efficient neural networks (Han et al., 2015b), we propose find better local optima by starting with a NN with many hidden neurons and then use connection pruning with an annealing schedule to achieve both feature and node selection and obtain a well trained NN. The advantage of using a annealing schedule with an iterative pruning approach is that we can control how many connections we want to keep in the end, making it easy to determine the minimum number of connections needed to reproduce the performance of the original unpruned network.

Figure 5 illustrates the pruning framework: we first train a fully connected NN with many hidden nodes to reach a local optimum, then we start connection pruning at that point with an annealing schedule $M_e$,
$$M_e = k + (p - k) \max\left(0, \frac{(N^{epoch} - N^{pretrain}) - 2e}{2e\mu + (N^{iter} - N^{pretrain})}\right) \tag{2}$$
where $k$ is the target number of connections desired to be kept at the end, $e$ is the current training iteration number, and $\mu$ is a parameter that controls the annealing rate.

---

**Algorithm 1 Connection Pruning with Normalization and Annealing (CPNA)**

**Input:** Training set $T = \{(\mathbf{x}_i, y_i) \in \mathbb{R}^p \times \mathbb{R}\}_{i=1}^n$, initial number of hidden neurons $H^l$, target number of weight connections $m_h^l$, pre-train iterations $N^{pretrain}$, total training iterations $N^{total}$, annealing schedule $M_e, e = 1, .., N^{total}$.
**Output:** Trained NN with $m_h^l$ weight connections in layer $l$.

1: Initialize a NN with $H$ hidden neurons with random initialization
2: **for** $e = 1$ to $N^{iter}$ **do**
3:    **if** $e > N^{pretrain}$ **then**
4:      **for** $l = 1$ to $L$ **do**
5:        Normalize hidden node $j$:
$$\mathbf{w}^{l+1} \leftarrow \|\mathbf{w}_j^l\| \mathbf{w}^{l+1}, b_j^l \leftarrow \frac{b_j^l}{\|\mathbf{w}_j^l\|}, \mathbf{w}_j^l \leftarrow \frac{\mathbf{w}_j^l}{\|\mathbf{w}_j^l\|} \tag{3}$$
6:        Remove weight connections to keep the $M_e$ connections with largest $|w|$
7:      **end for**
8:    **end if**
9:    Train the NN for the remaining iterations
10: **end for**

---

The connection pruning is done layer-wise. The hidden layer(s) and input layer can be pruned in different orders or even at same time, depending on which approach works best. Any connection that has been removed will not be reintroduced. If one of the hidden neurons has no effective connections to the neurons in the next layer left, we could remove it from the network since it no longer makes any contribution to the NN. If one of the input neuron (i.e data feature) has no effective connections to any hidden nodes, we can remove that feature for the feed-in data. After

pruning, the original fully connected neural network will generate a sparse neural network with only a few connections in each layer. We will keep training for a few more epochs so that the resulted sparse NN can eventually fall into a local optimum.

As the connection dropping criterion is based on the absolute magnitude of each weight connection in the NN, the weights need to be considered relative to the average of the activation of the input corresponding to each weight. We normalize the activation in each layer $l$ and incorporate the scaling factor into the weight $\mathbf{w}_j^{l+1}, j = 1, ..., h^{l+1}$, as follows. Observe that in Eq.(1), if we divide $\mathbf{w}_j^l$ and $b_j^l$ by a constant $c > 0$ and multiply $\mathbf{w}_k^{l+1}$ by the same $c$ we obtain an equivalent NN that has exactly the same output, due to the fact that we use ReLU activation. Since we normalized the data to the range $[-1, 1]^p$, we can equivalently scale the activation without using training data by normalizing the weight vectors $\mathbf{w}_j^l, j = 1, ..., h^l, l = 1, ..., L - 1$. The proposed detailed method for pruning the weight connections with annealing schedule including this normalization step is presented in Algorithm 1.

Observe that a similar algorithm can be obtained to directly remove NN hidden nodes instead of connections. We will call this algorithm also CPNA for simplicity.

To test the effectiveness of the weight normalization procedure from CPNA, we run the experimental results shown in Figure 6 for a single hidden layer neural network with 100 random initializations on the 5D XOR data with $p = 15, n = 3000$. For the CPA (connection pruning without weight normalization) and CPNA (connection pruning with weight normalization), we only perform connection pruning between the hidden layer and the output layer and leave the input layer connections untouched. We see both CPA and CPNA help improve the generalization capability of NNs, but the CPNA not only achieves the highest test AUC, but also is more consistent (the variance is smaller) than CPA in obtaining deeper local optima.

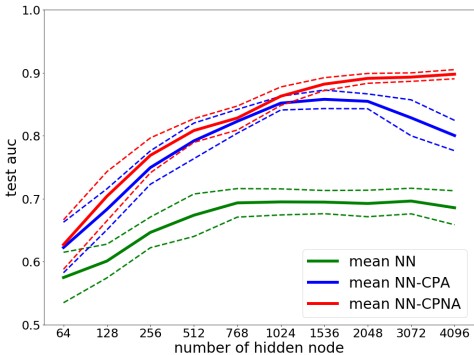

Figure 6: Average test AUC vs number of hidden nodes. The NN-CPA and NN-CPNA starting with $H$ hidden nodes as the NN, but are reduced to $64$ connections at the end of training. Shown are the means, and mean±std.

## 4 EXPERIMENTS

### 4.1 XOR RELATED SIMULATED DATASETS

In this section, we present experiments on 3D, 4D, 5D XOR datasets with a 3000 observations training set and 3000 different testing samples. The total number $p$ of features for each dataset will be selected such that the neural network almost lost all its generalization capability. For the 3D XOR dataset, we selected $p = 60$, for 4D XOR is $p = 40$ and 5D XOR is $p = 20$. Figure 3 shows that a one hidden layer NN without regularization can only obtain a test AUC close to 0.5 for these cases, which means it has nearly no generalization power.

We also train a one hidden layer neural network starting from a large number of hidden nodes, then gradually prune the weight connections to some target number, with the purpose of finding a deeper local optimum that can make the NN generalize better. We first perform connection pruning in the hidden layer (NN-CPNA-hidden), and leave the input connections which connect the features to the hidden neurons untouched. After that we prune connections to the input layer, which will be described below. After searching through a number combination of hyper-parameters, we obtained the best initial number of hidden nodes as 1024 for 3D XOR, 2048 for 4D XOR, and 4096 for 5D XOR, and the final number of connections as 12, 16 and 64 respectively. The obtained test AUC (Area Under the ROC curve) results using a default Adam optimizer and L2 penalty 0.001 with CPNA are displayed in Table 1.

| Test AUC(%) | | | | | |
|---|---|---|---|---|---|
| Dataset | NN(best) | NN-CPNA(hidden) | NN-CPNA(all) | NN-CPNA(random) | NN-CPNA(init) |
| 3D-XOR $p = 60$ | 82.53 | **96.58** | **99.58** | 70.47 | 71.68 |
| 4D-XOR $p = 40$ | 72.17 | **95.31** | **98.31** | 52.83 | 69.32 |
| 5D-XOR $p = 20$ | 66.24 | **80.03** | **96.83** | 53.21 | 66.29 |

Table 1: Comparison between the standard NN, pruned networks on the hidden layer and all layers, and retrained pruned networks with random weights and the initial weights from the beginning of CPNA.

In Table 1 are also displayed the best test AUC of a standard one hidden layer NN, the best result of a pruned sub-network that was retrained after the pruning stage using a random initialization, and the best result of a pruned sub-network that was retrained after the pruning stage using the initial weights used before pruning. Clearly, we see that our connection pruning technique that preserved the trained weights helps the NN have a far better generalization than the others networks, and the sub-network does not seem to find a good local optimum when retrained.

We further examine whether or not we can recover the performance of the sub-network after the pruned sub-network was obtained, either via retraining from scratch or from the corresponding initial original weights. In each case, we

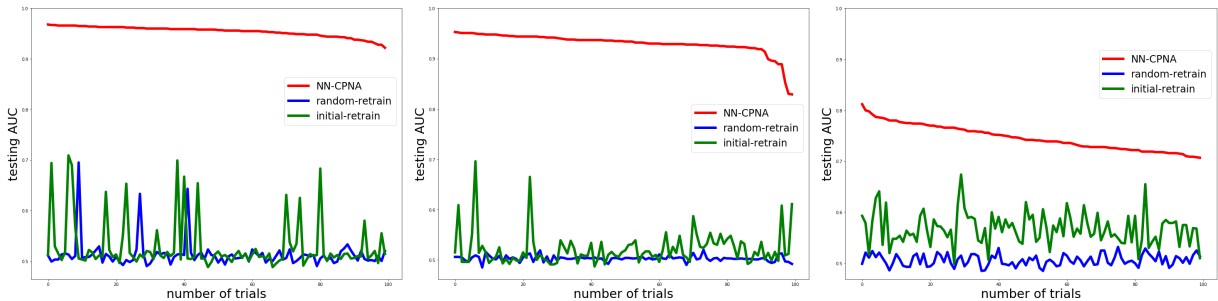

Figure 7: Testing AUC vs number of trials. Left: $k = 3, p = 60$; Middle: $k = 4, 40$; Right: $k = 5, p = 20$.

repeat 100 times the process of pretraining a fully connected neural network, applying CPNA to prune the hidden connections, and retraining the pruned sub-network using random initialization and initial weights, using the best combination of hyper parameters described above. We sorted the runs by the test AUc and displayed the obtained resulting curves in Figure 7.

From Figure 7 we see that for these three cases on the XOR data, neither retraining from scratch nor using the original initial weights can achieve the performance of the corresponding sub-network that was obtained by pruning the fully connected dense neural network. The retraining using initial weights seems works a little better than retraining from a random initialization, as sometimes the sub-network retrained with initial weights can obtain a test AUC beyond 0.6, but still way behind the sub-network obtained form large network pruning.

Besides pruning the connection between the hidden layer and output layer, we also prune connections to the input layer to remove irrelevant features, shown as NN-CPNA(all) in Table 1. This may further improve the performance for the XOR data. The obtained results of further pruning the input connections down to 24 for 3D, 32 FOR 4D, 256 for 5D XOR are also displayed in Table 1, and almost perfectly match the best result we could obtain from training the same XOR data without irrelevant features.

Figure 8 displays a complete training and test loss obtained during training a NN with CPNA on the 4D XOR data with $p = 40$. Also shown is the AUC evolution, which indicates that a good sub-network was obtained in the end, performing well on both training and testing data The fully connected neural network was first trained for 1200 iterations to reach a local optimum with a training AUC of 1.0 but a bad test AUC. After pruning the connections to the hidden nodes, the training AUC went back to 1 and the test AUC improved considerably. Finally, after pruning the connections to the input layer to remove irrelevant features, the test AUC obtains a comparable performance to the training AUC, which means this is a good deep local optimum. This figure is a good illustration of how the loss and AUC evolve during the training procedure. In most cases the number of training iterations can be reduced to have same final testing performance.

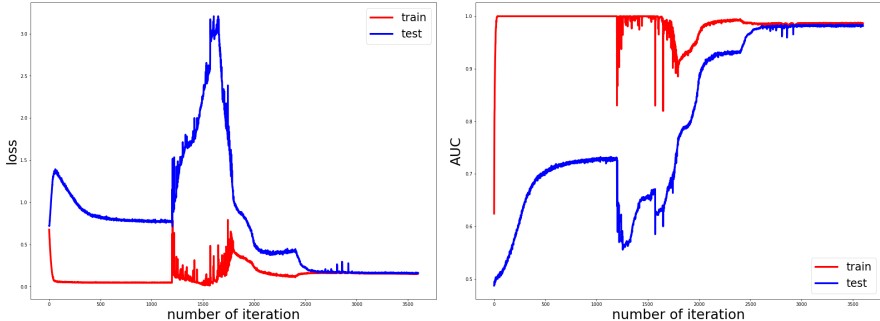

Figure 8: Loss and AUC evolution for training a pruned sub-network using CPNA for $k = 4, p = 40$ XOR data.

## 4.2 REAL DATASETS

In this section, we perform an evaluation on a number of real multi-class datasets to compare the performance of a fully connected NN and the sparse NN obtained by the CPNA Algorithm 1. The real datasets were carefully selected from the UCI ML repository (Dua & Graff, 2017) to ensure that the dataset is not large (the number of data samples less than 10000) and that a standard fully connected neural network (with one hidden layer) can have a reasonable generalization power on this data. If a dataset is large, then the loss landscape is simple and the neural network can find a deep local optimum easily, so there is no need for connection pruning to escape bad optima. If a dataset is such that a neural network can rarely be trained on it successfully, it means that the loss might not have any good local optima, then again connection pruning might not make sense. The details about the datasets are displayed in Table 2.

| Dataset | Number of classes | Number of features | Number of observations |
|---|---|---|---|
| Car Evaluation | 4 | 21 | 1728 |
| Image Segmentation | 7 | 19 | 2310 |
| Optical Recognition of Handwritten Digits | 10 | 64 | 5620 |
| Multiple Features | 10 | 216 | 2000 |
| ISOLET | 26 | 617 | 7797 |

Table 2: Datasets used for evaluating the performance of fully connected NN and sparse NN with CPNA.

Our real dataset experiments are not aimed at comparing the performance with other classification techniques, but to test the effectiveness of CPNA in guiding neural networks to find deeper and better local optima, we will combine all the samples including training, validation and testing data to form a single dataset for each data type first, and then divide them into a training and testing set with a ratio $4 : 1$. The obtained training dataset will be used in a 10-run averaged 5-fold cross-validation grid search training process to find the best hyper-parameter settings of a one hidden layer fully connected neural network. After getting the best hyper-parameter setting from the cross-validation, we use them to retrain the fully connected NNs with the entire training dataset 10 different times, and each time we record the best test accuracy. This procedure is used for the fully connected NN, and the NN with CPNA with different sparsity level and record the best sparsity level and testing accuracy. Finally, we will also train a so-called "equivalent" fully connected neural network with roughly the same number of connections as the best sparse neural network we get from CPNA.

| | NN(best) | NN(equivalent) | NN+CNPA |
|---|---|---|---|
| Car Evaluation Dataset | | | |
| Number of Connections | 21x64+64x4 = 1600 | 21x6+6x4 = 150 | 120+32 = 152 |
| Test Accuracy | **100.0±0.00** | 98.23±0.06 | **100.0±0.00** |
| Image Segmentation Dataset | | | |
| Number of Connections | 19x256+256x7 = 6656 | 19x14+14x7 = 364 | 266+98 = 364 |
| Test Accuracy | 96.87±0.72 | 96.27±0.58 | **98.40±0.32** |
| Optical Recognition of Handwritten Digits Dataset | | | |
| Number of Connections | 64x512+512x10 = 37888 | 64x27+27x10 = 1998 | 1792+160 = 1952 |
| Test Accuracy | 98.80±0.29 | 98.25±0.19 | **99.01±0.20** |
| Multiple Features Dataset | | | |
| Number of Connections | 216x64+64x10= 14464 | 216x4+4x10 = 904 | 583+320 = 903 |
| Test Accuracy | 97.85±0.80 | 95.45±0.98 | **98.15±0.82** |
| ISOLET Dataset | | | |
| Number of Connections | 617x64+64x26= 41152 | 617x9+9x26 = 5787 | 4683+1118 = 5801 |
| Test Accuracy | 96.73±0.50 | 94.31±0.61 | **96.91±0.54** |

Table 3: Performance results of NN(best), NN(equivalent) and NN+CNPA for each dataset.

All the experiments were trained with the default Adam optimizer (Kingma & Ba, 2014), the number of hidden nodes was searched in $\{16, 32, 64, 128, 256, 512\}$, the L2 regularization coefficient was searched in $\{0.0001, 0.001, 0.01, 0.1\}$, the batch size was searched in $\{16, 32, 64\}$. Other NN training techniques like Dropout (Srivastava et al., 2014) and Batch Normalization (Ioffe & Szegedy, 2015) were not used in our experiments due to the simplicity of the architecture of experimented NNs. The comparison result are listed in Table 3.

We see from Table 3 that using CPNA to guide the search for a local optimum leads to NNs with good generalization on all these datasets, easily outperforming a NN of an equivalent size (with a similar number of connections) and in most cases even the standard NN with the best generalization to unseen data. The experiments show that the XOR data is indeed an extreme example where deep local optima are be hard to find, but even these datasets exhibit some non-equivalent local optima and the things we learned from the XOR data carry over to these datasets to help us train NNs with better generalization.

## 5 CONCLUSIONS

This paper presented an empirical study of the trainability of neural networks and the connection between the amount of training data and the loss landscape. We observed that when the training data is large (where 'large" depends on the problem), the loss landscape is simple with a small number of deep local optima. When the training data is small, the number of local optima can become very large, making it hard to find deep local optima. For these cases we introduce a method for training a neural network that avoids many local optima by starting with a model with many hidden neurons and gradually removing weak connections to obtain a sparse network trained in a deep minimum. Moreover, the performance of the obtained pruned sub-network is hard to achieve by retraining using either random initialization or initial weights due to the existence of many shallow local optima around the deep minimum. Experiments also obtain good results on a number of real datasets using the same network pruning technique.

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

## A  APPENDIX

You may include other additional sections here.

