# OpenReview forum: "Finding Deep Local Optima Using Network Pruning"
_ICLR.cc/2020/Conference — Reject_

### Official Review · AnonReviewer1 · 2019-10-21
**Official Blind Review #1**

**Rating:** 3

**Review:**

This paper presents a few sets of experiments related to deep local optima. The first set of experiments are to study the local minima of two layer neural networks with ReLU activation for the hidden layer and specialized for XOR problem. The paper claims that it is quite easy to find a deep local minimum with good generalization when the number of irrelevant features is small, and it becomes harder to find a deep minimum with good generation as the number of irrelevant features increases. It also claims there is a large difference between the test AUC of the best local minimum and the worse one if the training data is difficult.

The second experiment set is about pruning fully connected neural networks to find deeper and better optima. The proposed pruning method employs a annealing schedule and iteratively pruning connections to reduce features and nodes. For XOR datasets, the pruning seems be effective. For several real datasets, pruned models are better than original or equivalent models.

One thing concerns me is that there are a lot of experiment settings seem to be arbitrary. For instance, why use 500 hidden nodes, p is 4, 16, then 100, ...It will be better to explain why those setting are representative so the statements derived from those are valid.

For Figure 1 and 2, why switch sequence? The top 3 subfigures in Figure 1 is AUC, but the top 3 subfigures of Figure 2 is Loss. It is a bit confusing.

The paper is interesting and the experiments are comprehensive. I think the results and conclusion are specific for FC networks. It will be more interesting to study on CNN, etc.  Overall, I am a bit concerned with the significance of this paper.




**Experience Assessment:**

I have read many papers in this area.

**Review Assessment: Checking Correctness Of Derivations And Theory:**

I assessed the sensibility of the derivations and theory.

**Review Assessment: Checking Correctness Of Experiments:**

I carefully checked the experiments.

**Review Assessment: Thoroughness In Paper Reading:**

I read the paper at least twice and used my best judgement in assessing the paper.

---

### Official Review · AnonReviewer3 · 2019-10-24
**Official Blind Review #3**

**Rating:** 3

**Review:**

This submission studies losses at local minima of a set of neural networks trained on an XOR-like synthetic dataset, finds that local minima are of varying quality, and proposes a network pruning method to find better local minima. The pruning method is evaluated on XOR-like datasets as well as real-world datasets.

The use of an XOR-like dataset to study loss landscapes is interesting, making for a controlled and analyzable setting to carry out the study. The way the authors set it up, the XOR-like problem involves nuisance variables that naturally introduce suboptimal local minima into the loss landscape (this is my observation as a reviewer -- I am not sure if the authors were aware of this). I am unsure if Section 2 of the paper was intended as a core contribution or as a motivation for the pruning algorithm proposed in Section 3. Given the set-up’s simplicity, a short theoretical argument (maybe even a theorem) about the quality and number of local minima one would expect to find could have been more concise and compelling than the empirical analysis from the paper. The findings from Section 2 may not be surprising enough to warrant two full pages.

Section 3 proposes a network pruning method to find better local minima. The authors cite a paper by Adrian Barbu as the inspiration for their pruning algorithm with annealing, and use it “to improve the capability of NNs to find a deep local minimum even when there are irrelevant variables”. The cited paper by Barbu as well as https://arxiv.org/pdf/1805.01930.pdf (also by Adrian Barbu, not cited, maybe because it appeared) explore feature selection and regularization with (nearly) the same annealed pruning algorithm in some detail. I would be grateful if the authors could highlight the differences between their work and Barbu’s.

I vote to “weak reject” this paper. The paper discusses interesting ideas, but other ICLR submissions present deeper and more novel material, and there appears to be some (unintentional, I believe) overlap with already-published work. I recommend that the authors cite and discuss https://arxiv.org/pdf/1805.01930.pdf , and possibly submit the paper at a less competitive conference.


Further comments / questions / advice
=================================

- It would be helpful if the authors made more clear what they consider the key contributions of their paper. If contributions build directly on earlier work, it’s helpful to highlight the differences.

- Section 4.2 states that datasets were “carefully selected” in what sounds like a case-by-case basis, probably with the goal of finding data sets on which CPNA outperforms networks trained with vanilla gradient descent methods. This process would have selection bias and surface data sets on which CPNA outperforms. I could be grateful if the authors could clarify if this was indeed the process, or if a less biased criterion was used. For example, one could have chosen data sets on which a 1-layer fully connected neural network achieves between 50% and 90% F-1.

- A reader of the paper might wonder for what data sets they should use CPNA in order to train network that achieves low out-of-sample loss. I could be grateful if the authors could comment on this. Following up on the previous point: it would be great the authors could include data sets where CPNA does not outperform.

- Could the authors include information on how long training takes for the experiments from Table 3?

- https://openreview.net/pdf?id=HkghWScuoQ should probably be cited

- https://arxiv.org/pdf/1805.01930.pdf should definitely be cited

**Experience Assessment:**

I have read many papers in this area.

**Review Assessment: Checking Correctness Of Derivations And Theory:**

I carefully checked the derivations and theory.

**Review Assessment: Checking Correctness Of Experiments:**

I assessed the sensibility of the experiments.

**Review Assessment: Thoroughness In Paper Reading:**

I read the paper thoroughly.

---

### Official Review · AnonReviewer2 · 2019-10-24
**Official Blind Review #2**

**Rating:** 6

**Review:**

The paper addresses the very important topic of local minima in Deep Learning. This is one of the central questions in the theory of Deep Learning for the last years, and despite many interesting results the main questions remain wide open.
The reviewer really likes the approach proposed in the paper, to use a simple model and an artificially generated data to study a certain phenomenon. The reviewer represents the opinion that more focus on such setups would greatly benefit the community in terms of progressing the theoretical understanding.
The claim made in the paper that there is a relationship between the number/suboptimality of local minima  and the scarcity of the data is both convincing and interesting. The result is well motivated and explained.
What the reviewer thinks the paper would greatly benefit from would be improving the Related Work section. There was a lot of valuable work in the field done in the past years that ids very relevant to the results presented that is not mentioned.

**Experience Assessment:**

I have published in this field for several years.

**Review Assessment: Checking Correctness Of Derivations And Theory:**

I assessed the sensibility of the derivations and theory.

**Review Assessment: Checking Correctness Of Experiments:**

I assessed the sensibility of the experiments.

**Review Assessment: Thoroughness In Paper Reading:**

I read the paper at least twice and used my best judgement in assessing the paper.

---

### Decision · Program_Chairs · 2019-12-19

**Decision:**

Reject

**Comment:**

This paper provides empirical evidence on synthetic examples with a focus on understanding the relationship between the number of “good” local minima and number of irrelevant features. The reviewers find the problem discussed to be important. One of the reviewers has pointed out that the paper does not present deep insights and is more suitable for workshops. The authors did not provide a rebuttal, and it appears that the reviewers opinion has not changed.

The current score is clearly not sufficient to accept this paper in its current form. Due to this reason, I recommend to reject this paper.